# Perisomatic GABAergic synapses of basket cells effectively control principal neuron activity in amygdala networks

**Judit M Veres[1,2], Gergő A Nagy[1], Norbert Hájos[1]\***

[1]'Lendület' Laboratory of Network Neurophysiology, Institute of Experimental Medicine, Hungarian Academy of Sciences, Budapest, Hungary; [2]János Szentágothai School of Neurosciences, Semmelweis University, Budapest, Hungary

**Abstract** Efficient control of principal neuron firing by basket cells is critical for information processing in cortical microcircuits, however, the relative contribution of their perisomatic and dendritic synapses to spike inhibition is still unknown. Using in vitro electrophysiological paired recordings we reveal that in the mouse basal amygdala cholecystokinin- and parvalbumin-containing basket cells provide equally potent control of principal neuron spiking. We performed pharmacological manipulations, light and electron microscopic investigations to show that, although basket cells innervate the entire somato-denditic membrane surface of principal neurons, the spike controlling effect is achieved primarily via the minority of synapses targeting the perisomatic region. As the innervation patterns of individual basket cells on their different postsynaptic partners show high variability, the impact of inhibitory control accomplished by single basket cells is also variable. Our results show that both basket cell types can powerfully regulate the activity in amygdala networks predominantly via their perisomatic synapses.

**\*For correspondence:** hajos@koki.hu

**Competing interests:** The authors declare that no competing interests exist.

## Introduction

Perisomatic region of cortical principal neurons (PNs), comprising the cell body, proximal dendrites and the axon initial segment, is a subcellular domain that is a critical site for the input integration and for the generation of the action potential, the output of neurons. Inhibitory inputs arriving onto this region are in a position to powerfully regulate the firing of individual PNs and to synchronize the activity of large neuronal ensembles, leading to an efficient control of neural processing within local networks (*Buhl et al., 1994*; *Cobb et al., 1995*; *Miles et al., 1996*). Two major types of GABAergic cells are known to innervate the perisomatic region in all cortical areas: axo-axonic cells form synapses exclusively on the axon initial segment of PNs, while basket cells (BCs) target the soma and proximal dendrites (*Somogyi et al., 1983*; *Somogyi, 1977*; *Veres et al., 2014*; *Gulyás et al., 1993*). The latter interneuron type is usually divided into two groups by the mutually exclusive expression of neurochemical markers: the calcium binding protein parvalbumin (PV) and the neuropeptide cholecystokinin (CCK). By having distinct functional characteristics, the two types of basket cells have been proposed to contribute differently to network operations (*Armstrong and Soltesz, 2012*; *Bartos and Elgueta, 2012*; *Freund and Katona, 2007*). Parvalbumin-containing basket cells (PVBCs) play a primary role in controlling and synchronizing the firing of PNs in large ensembles during distinct brain states (*Sohal et al., 2009*; *Stark et al., 2014*; *Schlingloff et al., 2014*; *Gulyás et al., 2010*; *Tukker et al., 2013*) as well as in gain control of sensory processes (*Atallah et al., 2012*; *Lee et al., 2013*). In contrast, due to the lack of selective genetic tools, much less is known about the function of cholecystokinin-expressing basket cells (CCKBCs). In an influential review it has been proposed that these GABAergic cells may provide a modulatory effect on network dynamics

conveying information from subcortical areas about emotional, motivational and general physiological states (*Freund and Katona, 2007*), a hypothesis that has not been challenged yet.

In the hippocampus, both PVBCs and CCKBCs preferentially innervate the soma and proximal dendrites of pyramidal cells (*Halasy et al., 1996*; *Gulyás et al., 1993*; *Földy et al., 2010*; *Szabó et al., 2014*). Their dendritic shafts, however, are targeted by different interneurons expressing PV or CCK, whose axonal arbor largely avoids the pyramidal cell layer (*Buhl et al., 1994*; *Halasy et al., 1996*; *Cope et al., 2002*; *Szabó et al., 2014*). The structural segregation of axon clouds of PV- and CCK-expressing GABAergic cells targeting distinct functional (perisomatic vs. dendritic) domains of pyramidal cells secures the division of labor between perisomatic and dendrite-targeting interneurons in the hippocampus. Perisomatic inhibitory interneurons effectively control the output, the spiking of pyramidal cells, while the dendritic interneurons regulate their input properties (*Miles et al., 1996*; *Pouille et al., 2013*; *Müllner et al., 2015*). This clear structural arrangement in axonal projections of interneurons is lacking in other cortical regions, and single BCs can innervate both the perisomatic region of PNs as well as their more distal dendrites (*Jiang et al., 2015*; *Kubota et al., 2015*; *Kawaguchi and Kubota, 1998*). This latter finding, however, has been demonstrated so far only for PVBCs (*Kubota et al., 2015*). These results obtained in neocortical areas raise three major questions: (i) is the potency of PVBCs and CCKBCs to regulate PN spiking similar or distinct, (ii) what is the relative contribution of perisomatic vs. dendritic synapses of BCs to the control of PN spiking, and (iii) is the innervation pattern and therefore the inhibitory efficacy of single BCs uniform on their different postsynaptic partners?

We have addressed these questions in the basal (basolateral) nucleus of the amygdala (BA), a cortical structure that plays a crucial role in fear learning (*Armony et al., 1995*; *Herry et al., 2008*; *Pape and Pare, 2010*). Previous studies established that similar types of interneurons can be found in this region as in other cortical regions, including the two types of BCs as well as axo-axonic cells (*Katona et al., 2001*; *McDonald and Betette, 2001*; *Sah et al., 2003*; *Spampanato et al., 2011*; *Bienvenu et al., 2012*; *Vereczki et al., 2016*). Using in vitro paired recordings, pharmacological manipulations and correlated light and electron microscopy we found that the two BC types could inhibit PNs with the same efficacy and innervated the same somato-dendritic membrane surface of their postsynaptic partners. Besides innervating the perisomatic region, both BC types also formed many synaptic contacts on distal dendrites at various ratios, however, the main determinant of the efficacy of their spike inhibition was the number of the synaptic contacts targeting the perisomatic region.

## Results

### CCKBCs and PVBCs provide inhibitory input onto PNs with similar magnitude

To compare the basic properties of inhibitory connections provided by the two BC types, interneuron-PN paired recordings were carried out using slices prepared from CCK-DsRed or PV-eGFP mice (*Figure 1A and C*). In the CCK-DsRed mouse strain, 84% of DsRed-expressing interneurons showed immunopositivity for CCK, and 92% of CCK-immunoreactive cells had DsRed expression (*Figure 1—figure supplement 1A*). In the PV-eGFP mouse strain, 95% of eGFP-expressing interneurons had immunolabeling for PV, and 92% of PV-immunostained neurons showed eGFP expression (*Figure 1—figure supplement 1B*). Quantification of the overlap between eGFP and DsRed expression in PV-eGFPxCCK-DsRed double transgenic mice (n = 2 mice) showed that only 2–3% of the cells expressed both fluorescent proteins (*Figure 1—figure supplement 1C*). These double labeled neurons had small soma size, dense local dendritic and axonal arborization as well as a firing character of late-spiking phenotype, features that together typify neurogliaform cells (*Figure 1—figure supplement 1D*) (*Mańko et al., 2012*). This interneuron type was excluded from further analysis.

We specifically studied BCs that were identified by neurochemical content and firing characteristics. The vast majority of the intracellularly labeled DsRed-expressing interneurons showed $CB_1$ immunoreactivity (97%, n = 30, *Figure 1B*), and all the tested PVBCs expressed a calcium binding protein calbindin (100%, n = 7, *Figure 1D*), as we reported previously (*Vereczki et al., 2016*). When tested with square current pulses, PVBCs showed the characteristic fast spiking phenotype, while CCKBCs displayed a slower, regular firing pattern in accordance with previous data (*Jasnow et al.,*

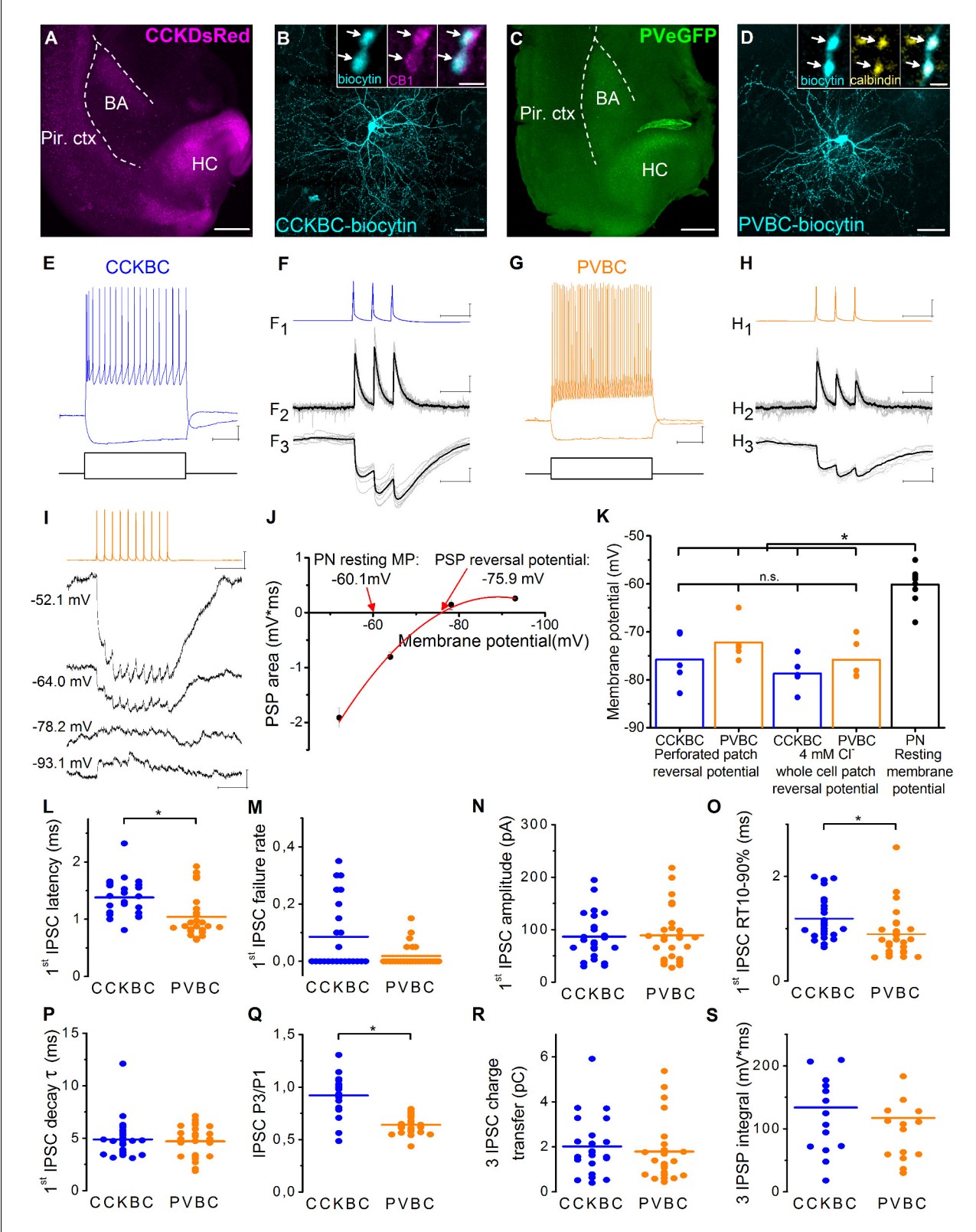

**Figure 1.** Similar magnitude and decay kinetics characterize the unitary inhibitory connections in CCKBC- and PVBC-PN pairs. (A, C) Images of horizontal slices taken from a CCK-DsRed and a PV-eGFP mouse, respectively, containing the basolateral amygdala region. Positions of the recorded BCs are shown in *Figure 1—figure supplement 2*. (B) Maximum z intensity projection image of a biocytin-filled CCKBC. Inset: Axon terminals of CCKBCs (blue) are immunopositive for type one cannabinoid receptor (CB1, magenta). (D) Maximum z intensity projection image of a biocytin-filled

*Figure 1 continued on next page*

*Figure 1 continued*

PVBC. Inset: Varicosities of PVBCs (blue) are immunopositive for calbindin. For more detailed neurochemical characterization of CCK-DsRed and PV-eGFP cells see *Figure 1—figure supplement 1*. (E and G) Representative voltage responses of a CCKBC (E, 400, −100 pA) and a PVBC (G, 300, −100 pA) upon step current injections. (F and H) Representative postsynaptic current (PSC, $F_2$, $H_2$) and postsynaptic potential (PSP, $F_3$, $H_3$) recordings in a CCKBC-PN (F) or in a PVBC-PN pair (H) in response to three action potentials evoked at 30 Hz ($F_1$, $H_1$). Ten superimposed consecutive traces in grey, averages in black. (I), PSPs recorded in a PN at different membrane potentials in gramicidin-based perforated patch recordings upon a train of 10 action potentials evoked at 40 Hz in a presynaptic PVBC. (J) The integral of the summed PSPs was calculated at each membrane potential and fitted with a second order polynomial curve to determine the reversal potential of the responses. (K) Comparison of the estimated reversal potential of PSP recorded in perforated patch and whole-cell patch-clamp mode using 4 mM Cl⁻ intrapipette solution and the resting membrane potential of PNs revealed the inhibitory nature of the BC connections (CCKBC: perforated patch mode: −75.74 ± 2.44 mV, whole cell mode: −78.67 ± 1.55 mV, p=0.296, n = 5–5 pairs; PVBC: perforated patch mode: −72.17 ± 1.88 mV; whole cell mode: −75.79 ± 1.90 mV, p=0.403, n = 5–5 pairs, both M-W test; PN resting membrane potential: −60.13 ± 1.09 mV). (L–S) Comparison of basic electrophysiological properties of IPSCs (L–R), and IPSPs (S) at the output synapses of CCKBCs and PVBCs recorded in whole-cell mode. Horizontal line represents mean. For data see *Figure 1—source data 1*. Scales: A and C: 0.5 mm; B and D: 50 µm, insets: 2 µm; E and G: 10 mV, 200 ms; F and H x: 50 ms, F1 and H1 y: 30 mV, F2 and H2 y: 50 pA, F3 and H3 y: 1 mV, I: 0.5 mV (PN), 30 mV (BC) and 100 ms.

The following source data and figure supplements are available for figure 1:

**Source data 1.** Basic electrophysiological properties of the output synapses of CCKBCs and PVBCs recorded in whole-cell mode.
**Figure supplement 1.** Characterization of the CCK-DsRed and PV-eGFP cells in the BA.
**Figure supplement 2.** Location of the presynaptic interneurons of the paired recordings in the BA.

*2009*; *Rainnie et al., 2006*; *Woodruff and Sah, 2007*) (*Figure 1E,G*). The synaptic coupling between the BCs and PNs was tested by evoking three action potentials in the presynaptic cell and measuring the spike-locked responses in the postsynaptic PN (*Figure 1F and H*). The probability to find a synaptically connected neighboring (<~150 µm) PN was high in both cases (85% for CCKBCs and 81% for PVBCs, respectively, n = 40 CCKBC and n = 37 PVBC).

To determine the reversal potential for Cl⁻ (an anion that primarily determines the magnitude of synaptic inhibition [*Bormann et al., 1987*; *Eccles et al., 1977*]) in the PNs at the output synapses of BCs, we first measured monosynaptic postsynaptic potentials (PSPs) at different membrane potentials in the postsynaptic neuron using perforated patch mode (*Figure 1I and J*, for details see Materials and methods). Then, we made the same measurements in whole-cell mode using 4 mM Cl⁻ containing intrapipette solution. The estimated reversal potentials of the PSPs recorded in perforated patch and whole-cell mode showed no significant difference (*Figure 1K*), therefore, in further experiments, we used the 4 mM Cl⁻ containing intrapipette solution. Furthermore, we found that the reversal potential of the PSPs arriving from both interneuron types were at more hyperpolarized values than the resting membrane potential of the PN, implying that the net effect of these BCs on their postsynaptic PN partners is inhibitory (p<0.001, Wilcoxon Signed Rank test, *Figure 1K*).

Next, we compared the physiological properties of the output synapses of the two BC types recorded in whole-cell voltage-clamp and current-clamp mode (*Figure 1L–S*, *Figure 1—source data 1*). Importantly, the latency of the signals indicated the monosynaptic nature of the connections from both cell types, however, IPSCs originating from PVBCs had significantly shorter latencies than those from CCKBCs. The failure rate, potency and amplitude of the first IPSC were not significantly different. The events arriving from PVBCs had faster rise time but similar decay time. In addition, there was a significant difference in the short-term plasticity of the signals: IPSCs originating from PVBCs onto PNs showed uniformly a marked depression, while those arising from CCKBCs were more variable and showed only a slight depression on average. Importantly, the charge transfer of the IPSCs and the area of IPSPs evoked by the two BCs in response to three action potentials at 30 Hz were not significantly different. These data revealed that some of the properties of the inhibitory connections originating from CCKBCs and PVBCs were rather similar, implicating that these interneurons may have the same potency to inhibit PN spiking.

## CCKBCs and PVBCs inhibit PN spiking with similar efficacy

To compare the ability of the two BC types to control the firing of their postsynaptic partners, we injected sinusoidal currents into the cell body of PNs in whole-cell mode to induce spiking, and the firing probability was measured with and without the presence of inhibitory input from a presynaptic BC (*Figure 2A*, for details see Materials and methods). When we compared the inhibitory efficacy of CCKBCs and PVBCs (i.e. 1- the probability of PN spiking in the presence of BC firing) we could not find significant differences. Both BC types could silence their postsynaptic partners with ~75% probability on average (*Figure 2C*). However, in both cell types there were BCs, which could veto PN firing with very high probability (>95% inhibitory efficacy, 42% of CCKBCs and 45% of PVBCs, n = 8 and 9, respectively), whereas others had only a weak inhibitory efficacy (<50% inhibitory efficacy, 21% of CCKBCs and 20% of PVBCs, n = 4 and 4, respectively). Those BCs that gave rise to IPSPs with larger area had a more pronounced inhibitory effect on their postsynaptic partner's spiking (*Figure 2D*). Although from the physiological point of view, the IPSP area is the most relevant factor determining the inhibitory efficacy of an interneuron, the low signal to noise ratio and the fluctuating membrane potentials make the postsynaptic potentials less precisely detectable in comparison to IPSCs obtained in voltage-clamp mode. Therefore, we used the charge transfer of IPSCs as the key electrophysiological property at a BC-PN connection, which correlated well with the IPSP area (*Figure 2E*) and similarly related to the inhibitory efficacy of BCs (*Figure 2F*).

In other set of experiments, we tested the impact of synaptic inhibition originating from BCs on the timing of PN spiking. We observed that IPSPs which failed to veto PN firing, could still alter the timing of the spike, delaying it by up to 38.5 ms (*Figure 2G–I*). The maximal delay caused by the two BC types was not significantly different (*Figure 2I*). We tested the importance of the timing of IPSPs by stimulating the presynaptic BC at different time points relative to the peak of the sinusoidal current evoking action potentials in the PN and measuring the caused delay in PN firing. We found that both cell types could control the firing of PNs in a ~110 ms-long time window (*Figure 2I*), with increased delaying effect in PN firing as the IPSP arrived closer to the sinusoidal cycle peak (i.e. to the theoretical time of the spike). These results together showed that inhibitory inputs from CCKBCs and PVBCs had the same powerful potential to postpone PN spiking.

To get an insight into the structural basis of the recorded BC-PN connections, the cells were loaded during the recordings with biocytin and a green Alexa dye (Alexa 488), respectively, to visualize them *post hoc*. With dual immunofluorescent labeling the pre- and postsynaptic cells were revealed in two different colors, enabling the detailed examination of the potential contact sites along the entire somato-dendritic surface of the postsynaptic PN using high resolution 3D confocal microscopy combined with 3D reconstruction of the postsynaptic cell for quantitative analysis (*Figure 3A–D*, *Figure 4A and B*). Correlated light and electron microscopic investigations showed that 85.4% of the potential contact sites (based on the confocal microscopic analysis) formed synaptic junctions on the soma (87.5%, n = 16 terminals from six pairs, *Figure 3E and G*) or dendrites (83.3%, n = 18 terminals from six pairs, *Figure 3F and G*) of the postsynaptic PNs (see detailed description of the analyzed contact numbers in *Figure 3—source data 1*). In addition, we found that the diameter of the targeted dendrites was variable, i.e. both cell types established synapses on thick (>1 μm, CCKBCs n = 10, PVBCs n = 6), presumably proximal and thin (<0.5 μm, CCKBCs n = 12, PVBCs n = 10), presumably distal dendritic segments at the same ratio (*Figure 3H*).

The analysis of the bouton distribution of 38 CCKBC-PN and 26 PVBC-PN pairs with confocal microscopy showed that the number of contacts is variable, ranging from 1 to 25 (*Figure 4*). The comparison of the target distribution of the two BC types along the PN membrane surface revealed no difference (*Figure 4C–E*), i.e. they targeted the same somato-dendritic compartment of PNs (*Figure 4E,K–S* test p=0.29), covering the whole length of the dendritic tree with exponentially decreasing density of contacts towards the tip of the dendrites (*Figure 4F*). We found that CCKBCs and PVBCs innervated their postsynaptic partners with similar number of boutons (*Figure 4G*), moreover, the average distance of the innervation along the dendritic tree (*Figure 4H*), the number of the contacts established on the perisomatic region (*Figure 4I*) and the ratio of perisomatic contacts among all boutons showed no significant difference (*Figure 4J*, all M-W test, p>0.05). Interestingly, there was only a small population of BCs, which restricted their innervation exclusively to the perisomatic region (i.e. ratio of perisomatic contacts = 1, 7.89% of CCKBCs and 3.84% of PVBCs, n = 3 and 1, respectively, *Figure 4J*), but most of the cells also contacted the dendrites with several

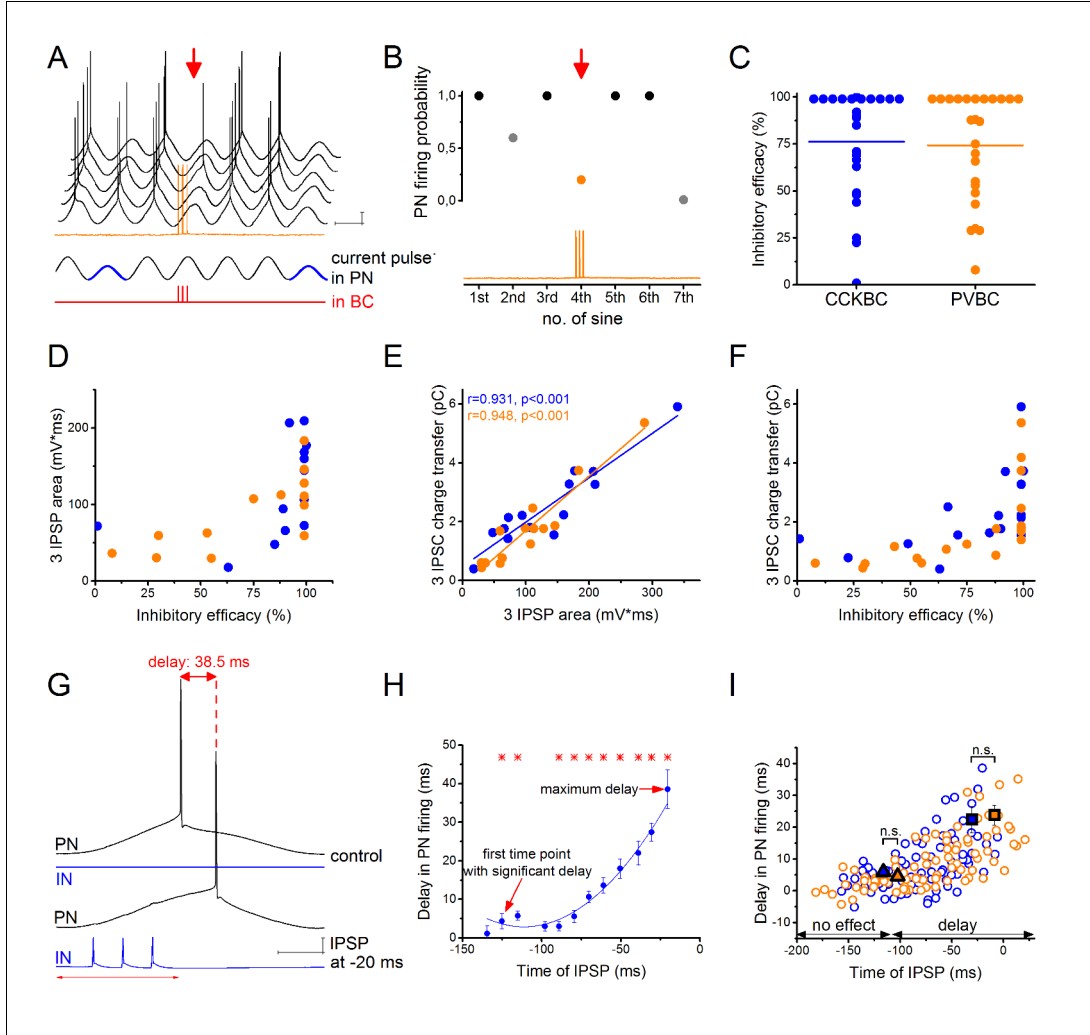

**Figure 2.** CCKBCs and PVBCs inhibit PN firing with similar efficacy. (A) Representative experiment for testing the capability of a BC to control spike generation in a PN. Sinusoidal current trains were injected into a PN at a theta frequency (3.53 Hz) to initiate firing and three action potentials were evoked at 30 Hz in the presynaptic PVBC 30–40 ms before the peak of the fourth cycle (for details see Materials and methods). Red arrow points to the cycle, when the presynaptic BC was stimulated. Voltage traces are offset for clarity. (B) Summary data of experiments shown in panel A: the firing probability of the PN was significantly suppressed within the cycle when the action potential train was evoked in the presynaptic PVBC. (C) Comparison of the inhibitory efficacy of CCKBCs and PVBCs. The inhibitory efficacy shows the probability of the suppression of action potential generation in the PN by BC firing (i.e.: 1- [PN firing probability during the control sinusoidal current cycles - firing probability when the BC is stimulated]) CCKBC: 76.2 ± 7.1%, n = 25, PVBC: 75.6 ± 6.5%, n = 25, p=0.88, M-W test. (D) The relationship between the inhibitory efficacy and the area of IPSPs. (E) The area of the IPSPs and the charge transfer of the IPSCs are strongly correlated. (F) The relationship between the inhibitory efficacy and IPSC charge transfer, indicating a larger inhibitory efficacy at a larger charge transfer. (G) Representative experiment for testing BCs ability to postpone PN firing. The timing of the evoked action potential in a BC was systematically shifted relative to the peak of the sinusoidal current injected into the PN. (H) The delay in PN firing as a function of the timing of synaptic inhibition in the pair shown in G. Asterisks indicate significant delay in firing compared to the peak of the cycle during the control period (paired sample Wilcoxon signed rank test p<0.05). (I) Pooled data from 7 CCKBC and 6 PVBC pairs. Squares show the average maximal delay (CCKBC: 22.40 ± 4.14 ms, n = 7, PVBC: 23.76 ± 3.16 ms, n = 6, p=0.72, M-W test) and triangles show the average last time point with significant delay in PN firing (CCKBC: −116.21 ± 14.47 ms, PVBC: −102.41 ± 18.58 ms, p=0.94, M-W test). Color codes are the same as in panel C (blue: CCKBCs, orange: PVBCs). Scales: A: 10 mV, 200 ms; G: 10 mV, 50 ms.

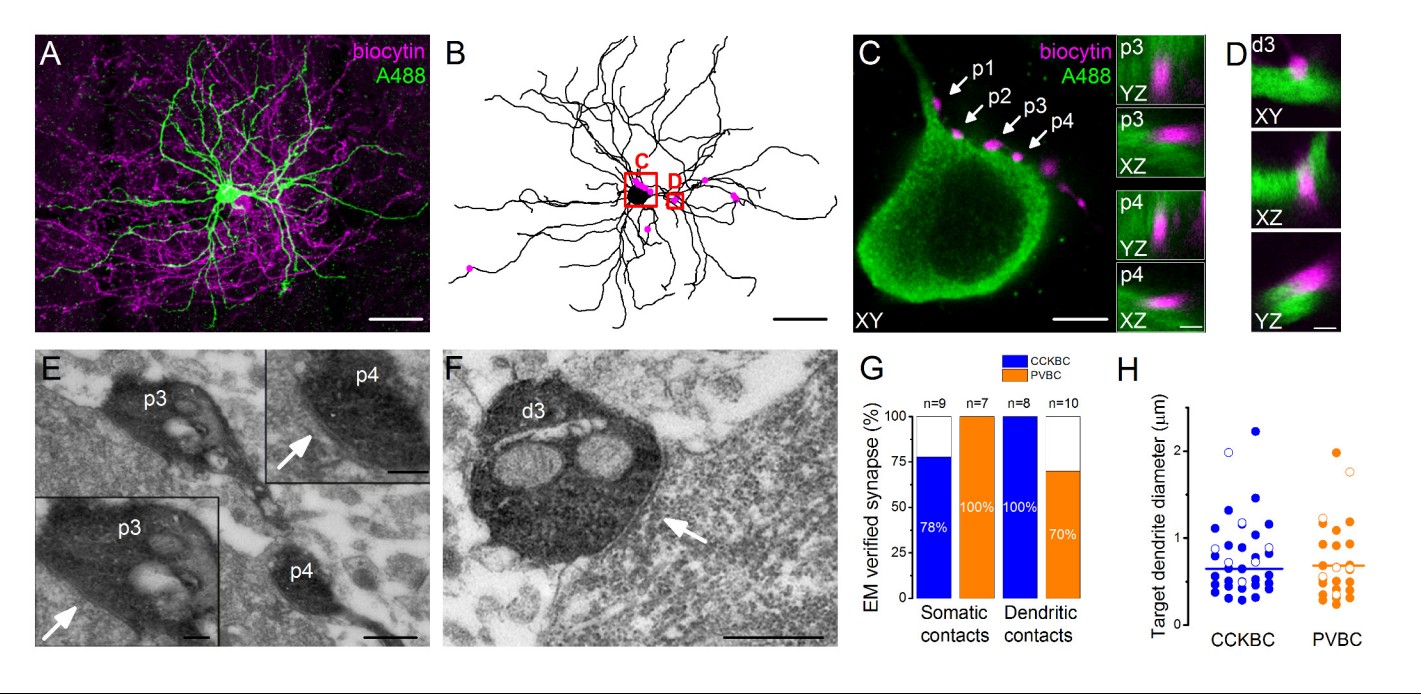

**Figure 3.** Morphological analysis of the synaptic connections in the recorded pairs. (A) Maximum z intensity projection of a 3D confocal image of a representative pair. Biocytin in the PVBC was visualized with streptavidin-conjugated Alexa 594 (magenta), while Alexa 488 in the PN is shown in green. For the differentiation of Alexa488 and eGFP signals see *Figure 3—figure supplement 1*. (B) Neurolucida reconstruction of the postsynaptic PN with the contact sites (magenta) originated from the PVBC. (C) High power magnification of the perisomatic region shown in B, which receives multiple contacts (arrows, perisomatic contact #1–4: p1-p4) from the presynaptic PVBC (magenta). Insets: 3D analysis of confocal images of p3 and p4 boutons shows close appositions of the pre- and postsynaptic structures. (D) High power magnification 3D confocal images of the region indicated in panel B, showing a close apposition between a dendrite-targeting bouton (d3) and a PN dendrite. (E) Electron micrographs of p3 and p4 shown in panel C. In both cases synaptic contacts with the soma of the postsynaptic PN (arrows) are visible. (F) An electron micrograph of d3 shown in panel D, confirming the presence of synaptic contact on the dendrite of the postsynaptic PN (arrow). (G) Ratio of close appositions observed at the confocal microscopic level forming synaptic contacts confirmed with electron microscopy. (H) Electron microscopic analysis of the diameters of the dendrites innervated by BCs shows that both BC types form synapses on dendrites with small and large diameters, respectively (CCKBCs: 0.79 ± 0.07 μm, n = 28, PVBC: 0.72 ± 0.08 μm, n = 36, p=0.63, K-S test). Open circles show contacts on the electrophysiologically recorded postsynaptic PNs, filled dots represent synapses on random targets of the biocytin-labeled varicosities of BCs in the sample. Scales (in μm): A and B: 50; C: 5, insets and D: 1; E: 0.5, insets: 0.2; F: 0.5. For the details of the analyzed contact numbers see *Figure 3—source datas 1* and *2*.

The following source data and figure supplement are available for figure 3:

**Source data 1.** Number of the pairs and analyzed contacts at the confocal microscopic level.
**Source data 2.** Summary table of the contacts analyzed at the electron microscopic level.
**Figure supplement 1.** Subcellular compartments of Alexa488 filled and subsequently immunolabeled cells are readily distinguishable from eGFP signal in PV cell somata.

synapses (92.11% of CCKBCs and 96.16% of PVBCs, n = 35 and 25, respectively). The number of perisomatic contacts was not dependent on the location of the postsynaptic cell in the BA (*Figure 5—figure supplement 1*), or on the distance between the pre- and postsynaptic cells (*Figure 5—figure supplement 2*), albeit the somata of neurons in paired recordings were close to each other (>150 μm), within a range where no large impact of the distance is expected (*Strüber et al., 2015*). Importantly, the ratio of the perisomatic region-targeting boutons was variable and showed a continuous distribution across the populations of both cell types. These results showed that CCKBCs and PVBCs innervate PNs with similar pattern, targeting somatic, proximal- and distal dendritic compartments with multiple synapses.

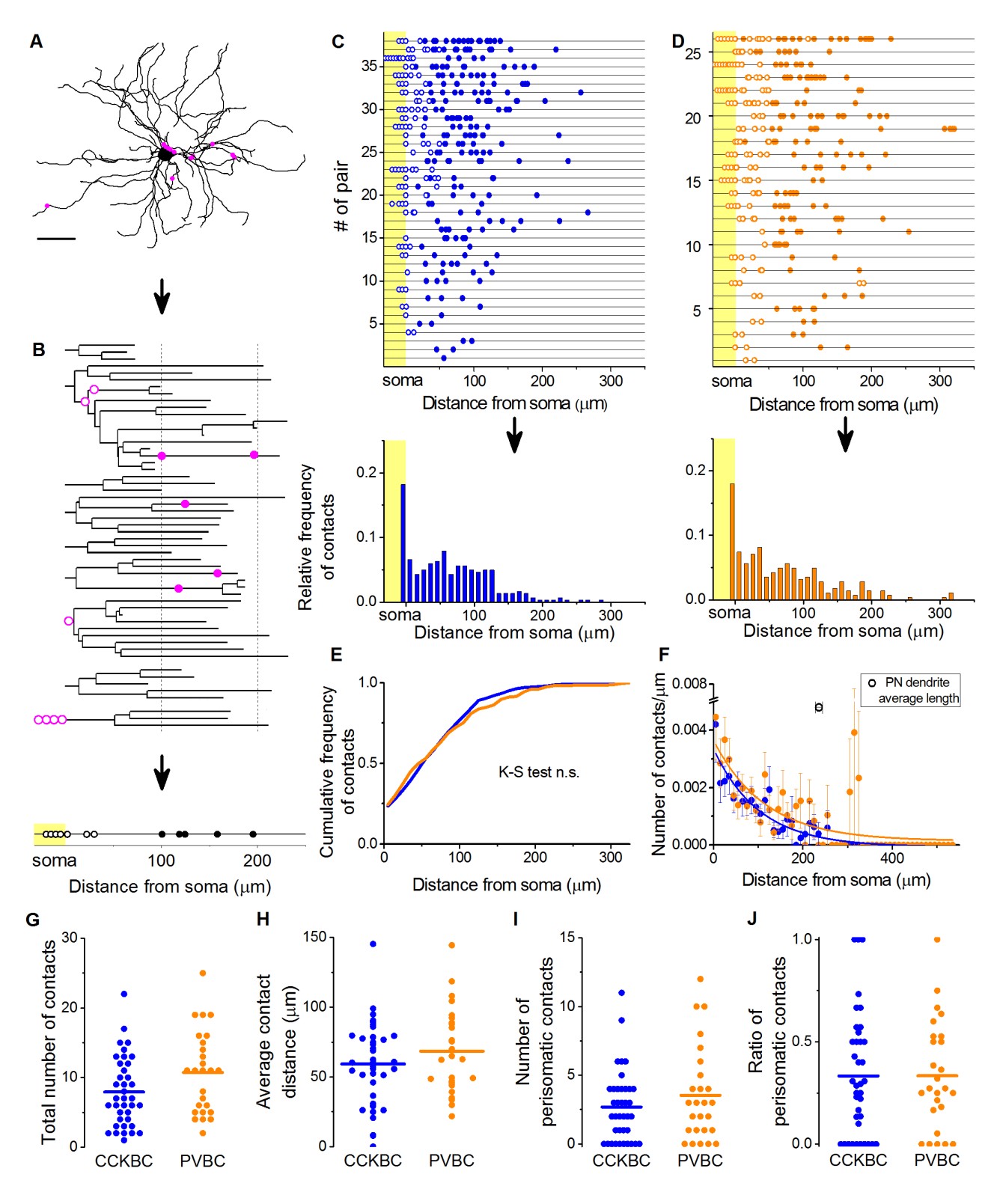

**Figure 4.** CCKBCs and PVBCs similarly innervate the soma-dendritic membrane surface of PNs. (**A** and **B**) Neurolucida reconstruction of the postsynaptic PN (same as in *Figure 3*) and its dendrogram analysis showing the contact sites on the perisomatic region (open magenta dots) and on the dendritic branches (filled magenta dots). Bottom on B: schematic visualization of the innervation pattern compressed along a single horizontal line. Somatic boutons shown in the yellow area, contacts on the perisomatic region are labeled with open dots and dendritic boutons with filled dots along

*Figure 4 continued on next page*

*Figure 4 continued*

the line indicating their distance from the soma. (**C** and **D**) Schematic innervation pattern of the reconstructed 38 CCKBC-PN and 26 PVBC-PN pairs. Bottom: Spatial distribution of the contacts, data from all pairs pooled together. (**E**) Comparison of the cumulative distribution of the contacts on PNs from the two BC types show no difference in the innervation patterns (p=0.29, K-S test). (**F**) BCs innervate the whole length of PN dendrites, however, the density of the contacts decreases exponentially towards the end of the dendrites (CCKBCs: r = 0.884, tau: 71.4 ± 8.2 µm; PVBCs: r = 0.557, tau: 79.7 ± 23.4 µm). Boutons contacting the cell bodies were excluded from this analysis. (**G–J**) Comparison of the total number of boutons (CCKBC: 7.92 ± 0.80, PVBC: 10.73 ± 1.16, p=0.065), the average bouton distance (CCKBC: 59.20 ± 4.80 µm, PVBC: 68.39 ± 5.97, p=0.38), the number of perisomatic boutons (CCKBC: 2.68 ± 0.41, PVBC: 3.53 ± 0.67, p=0.44) and ratio of perisomatic boutons (CCKBC: 0.33 ± 0.04, PVBC: 0.33 ± 0.05, p=0.83) of CCKBC-PN and PVBC-PN connections showed no significant difference (all M-W test, n = 38 CCKBC-PN pairs and n = 26 PVBC-PN pairs). Scale in A: 50 µm.

## The number of contacts targeting the perisomatic region determines the inhibitory efficacy

The above results raised an intriguing question. What is the contribution of perisomatic region- vs. dendrite-targeting boutons to control PN spike generation? Several lines of investigations have been performed to address this question. By analyzing the potential relationship between the electrophysiological properties of inhibitory connections and their underlying structural features, we found that the amplitude of the first IPSC is correlated with the number of the perisomatic boutons, but not with the total number of contacts (n = 20 CCKBC and 19 PVBC pairs, *Figure 5A–B*). The charge transfer of three IPSCs also showed higher correlation with the number of perisomatic contacts than with the total number of contacts (*Figure 5C and D*). Moreover, the inhibitory efficacy of both BCs may depend more on the number of perisomatic synapses than the total number of synapses as suggested by *Figure 5E–H*. The correlation of the electrophysiological and morphological data suggested that the most determining factor of the inhibitory efficacy is the number of the contacts established on the perisomatic region of PNs.

To directly test this assumption, we carried out whole-cell paired recordings combined with the pharmacological manipulation of GABAergic synaptic transmission locally at the perisomatic region by applying a GABA$_A$ receptor antagonist gabazine (1 µM). The size of the area affected by the gabazine application was determined in a separate experiment (*Figure 6A–C*) where we recorded and filled PNs with a fluorescent Alexa 488 dye and applied a small amount of GABA (100 µM) locally at different locations along the dendrites together with the somatic gabazine puffs (n = 4 cells, 5 dendrites on 22 locations). By defining the furthest location along the dendrites where somatic gabazine puffs still had a significant effect on the GABA-evoked IPSC amplitude (M-W test p<0.05), we estimated that the area affected by gabazine puffs has a radius of ~50 µm, which can largely cover the perisomatic region of PNs (*Figure 6C*) (*Vereczki et al., 2016*).

After establishing the appropriate conditions, we recorded synaptically connected BC-PN pairs and compared the IPSC charge transfer and inhibitory efficacy under control conditions and upon local gabazine application onto the perisomatic region (n = 4–4 CCKBC-PN and PVBC-PN pairs, *Figure 6D–J*). We found that both the IPSC charge and the inhibitory efficacy were significantly reduced when the perisomatic contacts were blocked (*Figure 6G,H*). It should be noted, however, that in pairs without perisomatic inhibitory inputs (either by lacking *a priori* or by functional elimination with gabazine application) we could still measure - albeit a significantly weaker - inhibitory effect (*Figure 6J*), implying that although perisomatic inputs are the major determinant in controlling PN spiking, yet a summation of sufficient numbers of dendritic inputs can also influence the output of the postsynaptic cell.

## Innervation patterns of single CCKBCs and PVBCs are variable on different postsynaptic cells

The analysis of the innervation patterns of both BC types on single postsynaptic PNs showed that some cells tend to target the soma and proximal dendrites (cells with high ratio of perisomatic contacts), while others prefer to target the dendrites (cells with low ratio of perisomatic contacts) (*Figure 4J*). As we showed earlier, the number of perisomatic contacts is the major determinant of the inhibitory efficacy of a BC (*Figure 6*), implying that BCs with different amount of perisomatic contacts may have a substantially different power to regulate PN activity. This raises the question

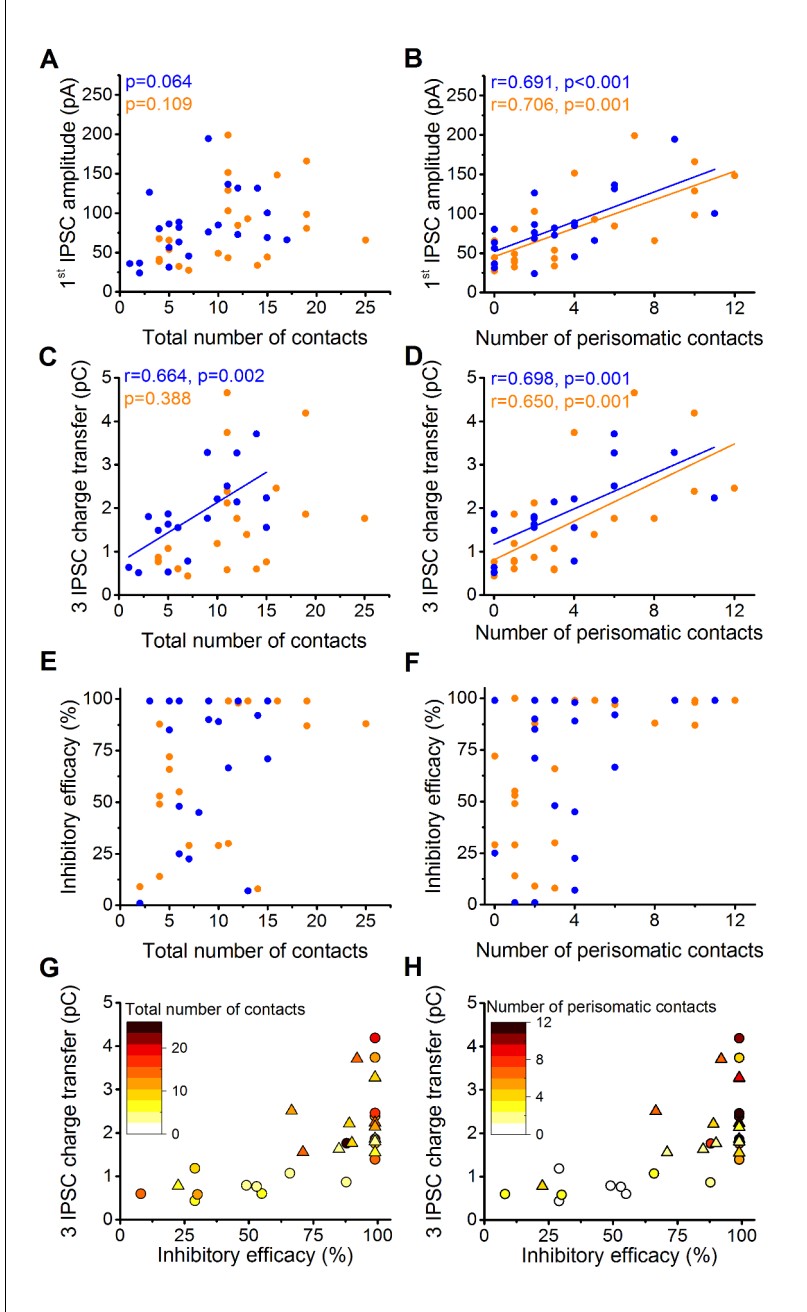

**Figure 5.** The magnitude of synaptic events and inhibitory efficacy show correlation with the number of the perisomatic contacts but not with the total number of contacts at individual connections. (A–F) Relationship between the structural and electrophysiological properties of the output synaptic contacts in CCK-PN (blue) and PVBC-PN (orange) pairs. Note that the IPSC amplitude, charge transfer and inhibitory efficacy show stronger correlation with the number of the perisomatic contacts than with the total number of contacts. Linear correlation is labeled with a solid line. Pearson's r values and significance levels of correlations are indicated in the top left corner of the plots. (G and H) Relationship between the inhibitory efficacy and IPSC charge transfer, plotted together with the total number (G) and number of perisomatic contacts (H) as a heat map showing that the inhibitory efficacy is rather dependent on the number of the perisomatic contacts than on the total number of contacts. On panel G and H triangles mark CCKBCs, circles label PVBCs. For additional analysis see *Figure 5— figure supplements 1,2*.

The following figure supplements are available for figure 5:

*Figure 5 continued on next page*

*Figure 5 continued*

**Figure supplement 1.** The number of contacts on the perisomatic region of the postsynaptic PN is independent of the location of the target cell within the BA.
**Figure supplement 2.** Properties of the connections are independent of the distance between the pre- and postsynaptic cells, at least below 150 μm.

whether, like in the hippocampus, there are BCs, which can be classified as 'classical' perisomatic region-targeting cells, because they innervate predominantly the perisomatic region of all of their postsynaptic partners, thereby potently controlling their spiking. Accordingly, other cells might be classified as dendrite-targeting interneurons, innervating mostly the dendritic shafts of PNs, having a less powerful effect on spike generation. Alternatively, a BC expressing PV or CCK could innervate some of their postsynaptic partners mainly at their perisomatic region, whereas other PNs could receive inputs from the same interneuron mainly on the dendrites, which would imply that a BC has a different inhibitory effect on its distinct postsynaptic partners. To address this question, we analyzed the target distribution of single biocytin-labeled BCs along the entire somato-dendritic membrane surface of three sequentially recorded and labeled postsynaptic PNs with the same method as used in the paired recordings (n = 8 CCKBC-PNs and 5 PVBC-PNs quadruplets, *Figure 7*). We found that in some cases the innervation patterns from one BC to three distinct PNs were similar, i.e. innervating mainly the perisomatic region (e.g. quadruplet #4 in *Figure 7D*) or more distal dendritic regions (e.g. quadruplet #8 in *Figure 7D*). However, there were some cases, where the same BCs innervated the soma of one postsynaptic PN with multiple contacts, whereas targeted only the dendrites of another PN (e.g. quadruplet #3 and 5 in *Figure 7E*). Similarly, some BCs innervated the perisomatic region of different postsynaptic PNs with similar number of terminals (e.g. quadruplet # one in *Figure 7D*, ranging from 0 to 2), while targeting others with variable number of terminals (e.g. quadruplet #3 in *Figure 7E*, ranging from 0 to 8). These data indicated that the innervation patterns of both CCKBCs and PVBCs could be highly variable and show a continuum in respect to the ratio of perisomatic contacts, if we evaluated the target distribution on multiple PNs.

To confirm and extend the conclusion of these latest investigations on a larger dataset, the fixed slices from paired recordings were immunostained against the voltage-gated potassium channel type 2.1 (Kv2.1), which labels the perisomatic region of the neurons (see Materials and methods and *Vereczki et al., 2016*). This approach allowed us to investigate the number and distribution of contacts from the presynaptic BCs both on a postsynaptic PN labeled with Alexa 488 in paired recordings and on the perisomatic region of 10–20 neighboring Kv2.1-immunolabeled cells (*Figure 8A,B*, n = 15 CCKBC-PN and 6 PVBC-PN pairs). Since the analysis of paired recordings showed no difference in the innervation patterns (*Figure 4E–J*), data form CCKBCs and PVBCs were pooled. We found that the number of perisomatic contacts from individual BCs was very variable on the Kv2.1-labeled profiles (ranging from 1 to 12 per Kv2.1-labeled cell, CV = 0.54 ± 0.03, *Figure 8C*), and showed no correlation with the number of perisomatic contacts on the intracellularly-labeled postsynaptic PN (*Figure 8D*). Moreover, from these datasets we could also determine the ratio of the contacts on Kv2.1-labeled perisomatic- and on unlabeled, presumably distal dendritic regions, thereby calculating the average perisomatic target ratio for single BCs at the population level (*Figure 8E*). The ratio of the contacts targeting the perisomatic region (i.e. on Kv2.1-labeled soma and proximal dendrites) was variable (CCKBCs: 48.6 ± 3.1%, range: 16.5–81.1%; PVBCs: 51.6 ± 3.5%, range: 17.4–84.9%) and formed a continuous distribution for both BC types (*Figure 8E*). We found no correlation between the ratio of perisomatic contacts obtained at the population level and those observed in the paired recordings (*Figure 8—figure supplement 1A*). However, the relationship between the average number of perisomatic contacts on Kv2.1-labeled profiles and the average ratio of perisomatic contacts at the population level (*Figure 8—figure supplement 1D*) implied that those cells that tend to target the perisomatic region at higher ratio at the population level, they also form more contacts on this membrane region of individual target cells (*Figure 8—figure supplement 1B–D*). This feature enables the BCs with high ratio of perisomatic contacts to control the spiking activity of the postsynaptic cells more effectively. These data together indicate that both CCKBCs and PVBCs tend to target the perisomatic region of PNs as a population,

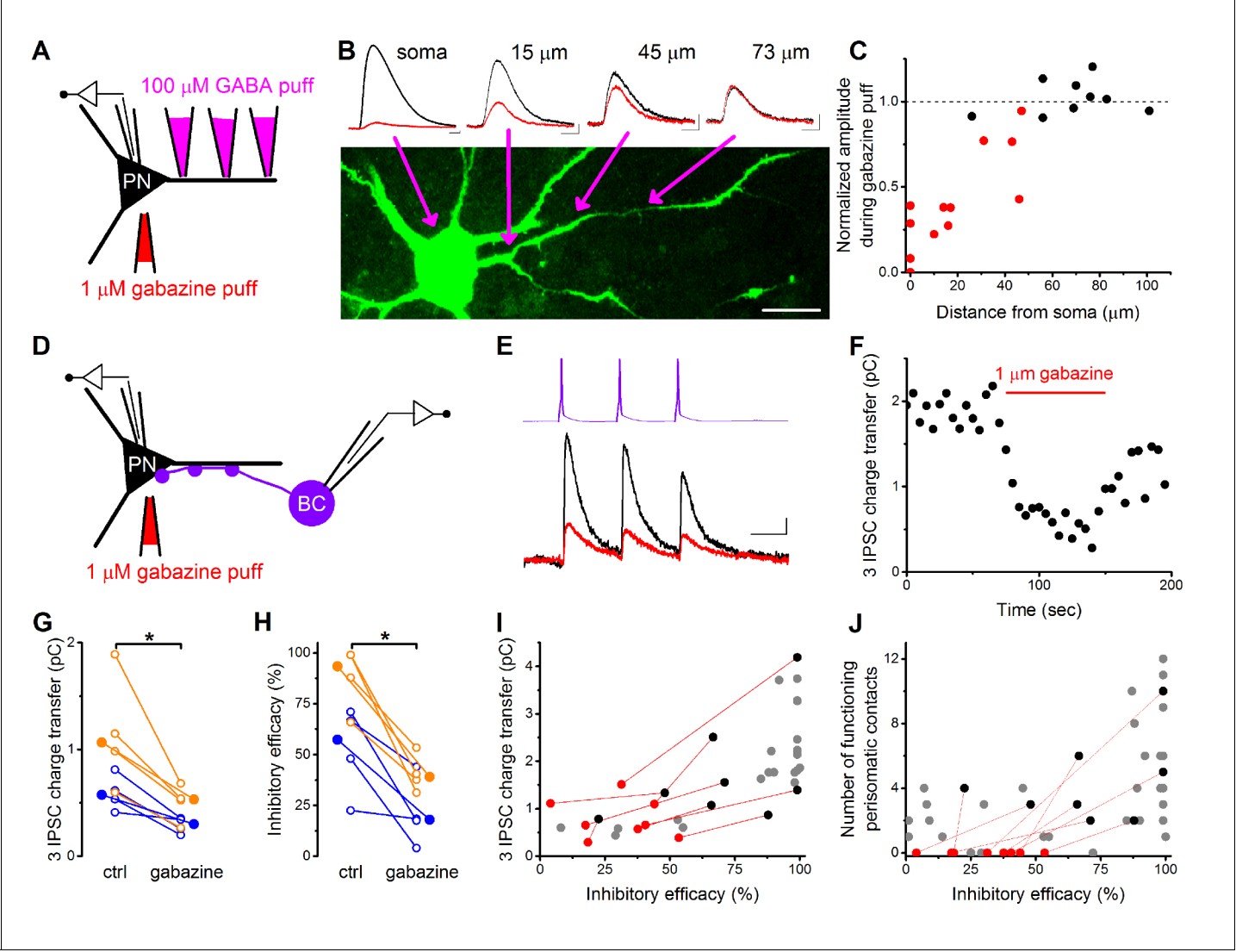

**Figure 6.** Pharmacological elimination of perisomatic contacts significantly reduces the inhibitory efficacy. (**A**) Schematic drawing of the experiment determining the area affected by local somatic gabazine application. (**B**) Representative recordings of the GABA puff evoked currents at various dendritic locations in control conditions (black) and upon somatic gabazine application (red). (**C**) Summary of recordings from 22 dendritic locations show that the somatic gabazine puff can significantly reduce GABAergic currents (red dots) in an area of 50 μm radius from the site of the application, which covers the perisomatic region. (**D**) Schematic drawing of the experiment measuring the effect of the elimination of perisomatic contacts by local somatic gabazine application. (**E** and **F**) Representative experiment showing the rapid and substantial drop in IPSC charge transfer upon gabazine application (applied for 75 s in panel F). (**G** and **H**) Elimination of perisomatic inhibitory inputs resulted in a significant suppression of the IPSC charge transfer and inhibitory efficacy (p=0.01 and p=0.007, respectively, Wilcoxon Signed Rank test). Blue: CCKBC-PN pairs, orange: PVBC-PN pairs. Open circles represent recorded pairs, filled circles represent median values. (**I** and **J**) Relationship between the inhibitory efficacy and IPSC charge transfer (**I**) or the number of functioning perisomatic contacts (**J**) before (black dots) and after (red dots) the pharmacological elimination of perisomatic inhibition (individual experiments connected with red line). Grey dots show data from previous recordings for comparison (data from *Figures 2F* and *5F*). Scales: B: 20 pA and 50 ms; 20 μm, E: 8 pA and 20 ms.

however, there is a marked variability in the number and ratio of the perisomatic contacts of individual BCs targeting different PNs, implying a variable inhibitory effect on distinct postsynaptic partners. Moreover, our data show that due to this variability it is not possible to determine the characteristic innervation pattern and inhibitory efficacy of a given BC based on the bouton distribution on a single postsynaptic partner, rather investigation of the target distribution at the population level is needed.

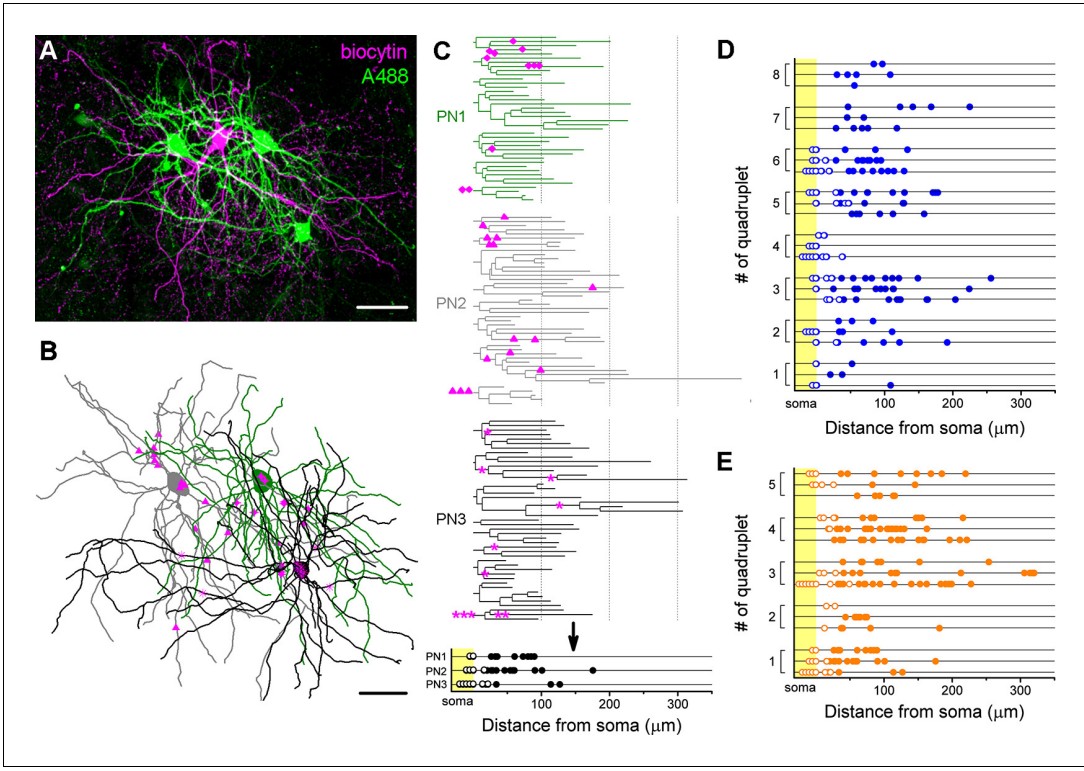

**Figure 7.** Target distribution of CCKBCs and PVBCs on multiple synaptic partners. (**A**) Maximum z intensity projection of a confocal image of a representative quadruplet. A single PVBC was filled with biocytin (visualized by Alexa 594, magenta) and three monosynaptically connected PNs (confirmed with electrophysiological recordings) were consecutively filled with Alexa 488 (green). (**B**) Neurolucida reconstruction of the three postsynaptic PNs (green) with the contact sites marked in magenta. Diamonds are contacts on PN1, triangles on PN2, stars on PN3. For the reconstructions the same method was used as in *Figure 4*. (**C**) On the dendrograms of the three reconstructed postsynaptic PNs the contact sites are marked. Bottom: innervation patterns of the presynaptic PVBC on the three postsynaptic PNs. (**D** and **E**) Innervation pattern of the reconstructed quadruplets (n = 8 CCKBC-PNs and 5 PVBC-PN quadruplets). Scales: A and B: 50 μm.

## Discussion

Our major findings are as follows: (1) CCKBCs and PVBCs provide equally potent inhibitory control of PN spiking in the BA. (2) Inhibitory synapses originating from the BCs cover the entire somato-dendritic membrane surface of PNs and only one third of the BC contacts is established on the peri-somatic region on average. (3) The number of the perisomatic contacts is the major factor determining the inhibitory efficacy on PN spiking. (4) The innervation pattern of individual BCs shows a high variability, therefore the impact of the inhibitory control provided by single BCs onto distinct post-synaptic partners can also significantly vary.

Our study is the first to provide a detailed comparative analysis of the effects of the two BC type outputs on PN spiking in a cortical structure. Analogous to what was found in the hippocampus (*Szabó et al., 2010*; *Hefft and Jonas, 2005*; *Glickfeld and Scanziani, 2006*), some characteristics of the output synapses of CCKBCs and PVBCs are slightly different, yet the efficacy of control these GABAergic cells exhibit on PN firing is surprisingly similar. Both cell types gave rise to synaptic currents with fast rise time, indicating the perisomatic origin of the postsynaptic events recorded in the PNs. Interestingly, although the rise time of the events originating from the CCKBCs was significantly slower, the location of their synapses was not distinct from those of PVBCs. This data indicates that the difference in rise times might derive from the spatiotemporal profile of neurotransmitter release, as it was described in the hippocampus (*Bucurenciu et al., 2008*; *Lenkey et al., 2015*). The distinct release machinery might also be responsible for the observed differences in the latency of the

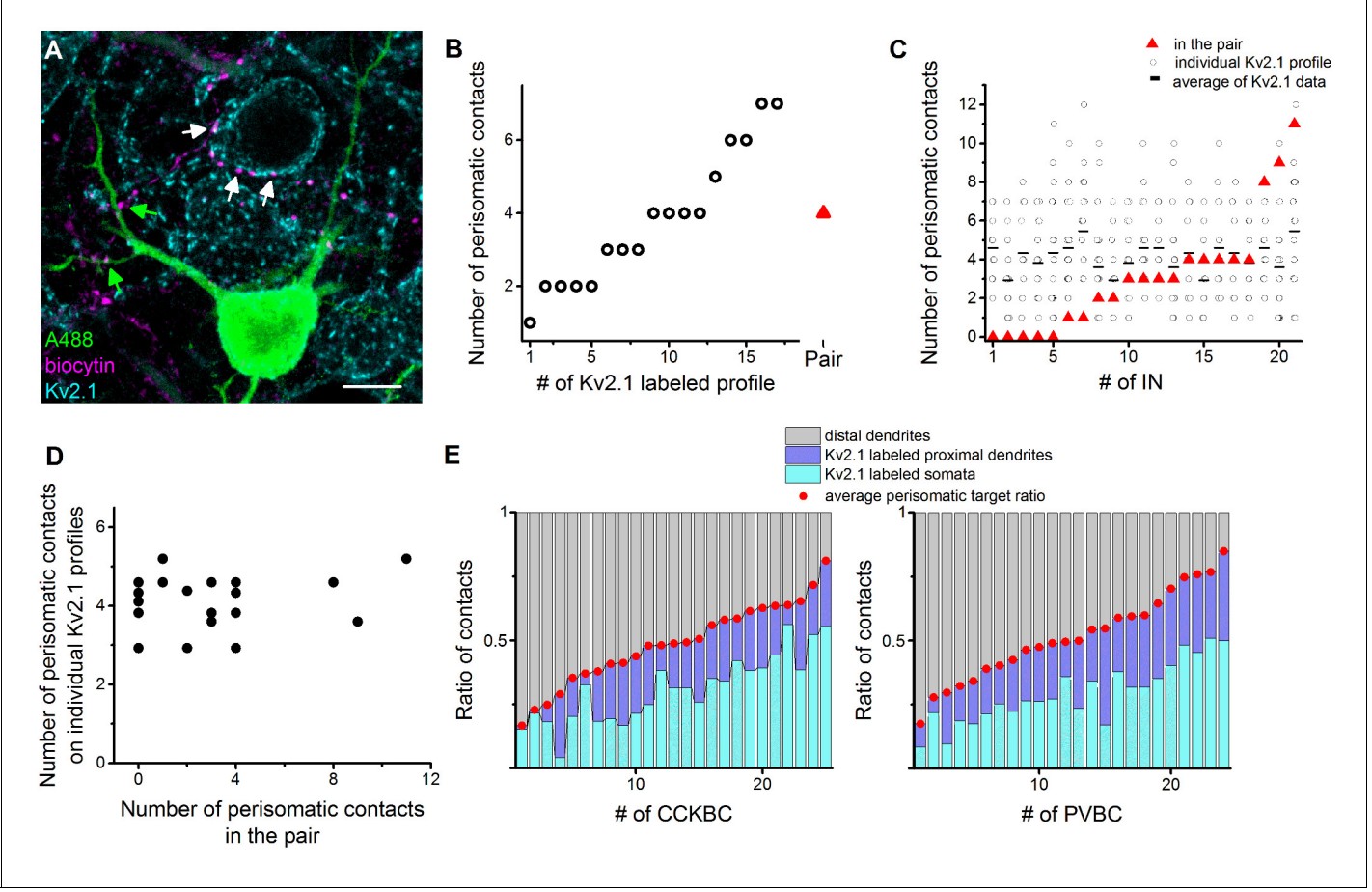

**Figure 8.** Comparison of the innervation patterns of CCKBCs and PVBCs at single-cell and population levels. (**A**) Kv2.1 immunostaining (blue) was used to label the perisomatic region of cells in slices where BC-PN pairs were visualized (biocytin in BC (magenta), and Alexa 488 in PN (green)). Contact sites from the same BC were identified on the intracellularly-labeled PN (green arrows) and on Kv2.1-expressing profiles (white arrows), enabling the investigation of the innervation pattern at both single-cell and population levels. (**B**) Analysis of the bouton distribution of a representative biocytin-labeled BC (shown in panel **A**) on 17 Kv2.1-labeled PNs. Data are arranged based on the number of biocytin-filled boutons contacting their perisomatic region (circles). For comparison, the number of the contacts formed on the intracellularly labeled postsynaptic PN is shown with a red triangle. (**C** and **D**) Summary data of the innervation patterns of 21 BCs showing that the number of the established contacts at the single-cell level (on the biocytin-filled postsynaptic PN) is independent of the average number of the contacts determined at population levels (i.e., on multiple Kv2.1-labeled PNs). (**D**) Data in C is arranged as a function of increasing contact number obtained in the pairs. (**E**) Target distribution analysis of CCKBCs and PVBCs at the population level obtained with Kv2.1 staining: ratio of contacts on the soma, proximal dendrites belonging to the perisomatic region and distal dendrites. Average perisomatic target ratio is shown with red dots. Data is arranged as a function of increasing ratio of contacts on the perisomatic region. Scale: 10 μm. For additional analysis see *Figure 8—figure supplement 1*.

The following figure supplement is available for figure 8:

**Figure supplement 1.** Possible innervation pattern strategies for PVBCs and CCKBCs.

responses. In our study, we tested the inhibitory efficacy by evoking three action potentials, which is an in vivo relevant activity pattern in the basolateral complex of the amygdala (*Bienvenu et al., 2012*; *Wolff et al., 2014*). Interestingly, this firing at 30 Hz induced a marked difference in the short-term plasticity of IPSCs at the output synapses of the two BC types, yet the IPSP areas recorded in the PNs had the same magnitude.

The dual color labeling of the recorded pairs made it possible to analyze the arrangement of the connections along the somato-dendritic membrane surface of PNs with an unprecedentedly high precision (*Kubota et al., 2015*). We found that both types of BCs innervated the perisomatic and the distal dendritic compartments up to 320 μm far from the soma (i.e. till the end of PN dendrites,

see also small caliber dendrite innervation by PVBCs reported in (*Bienvenu et al., 2012*). This innervation pattern resembles the properties of neocortical basket cells, as they also innervate distal dendritic regions of PNs with several synapses (*Tamás et al., 2000*; *Kubota et al., 2015*; *Kisvárday et al., 2002*). The functional implication of the presence of both somatic and dendritic inhibitory inputs from the same cell requires more investigations, however, BCs in the BA -besides controlling action potential generation and somatic signal integration (present study)- might be able to influence local dendritic computations by innervating distal dendritic segments (*Müllner et al., 2015*). Our results thus may suggest that in the neocortical and amygdalar microcircuits, in sharp contrast to that observed in the hippocampus (*Miles et al., 1996*), BCs might have a dual function, as their output synapses are in the position to control both the input and output features of PNs.

On average, ~30% of the synapses of both CCKBCs and PVBCs was found to target the perisomatic region, and both types of BCs formed comparable number of contacts on the perisomatic region (*Figure 4I*, 3–4 on average). Although a tendency could be observed that PVBCs might establish more contacts than CCKBCs, the inhibitory efficacy of these two cell populations was similar, which can be, at least partially, explained by the fact that the presence of multiple release sites at the axon terminals of CCKBCs in the BA (as in other cortical regions) is typical (*Katona et al., 2001*; *Yoshida et al., 2011*). In this study we used DAB/DAB-Ni labelling for EM level analysis, which didn't allow the recognition and separation of discrete release sites (*Biró et al., 2006*), thus the potential difference in the transmitter release site number between CCKBCs and PVBCs could not have been directly addressed.

The ratio of the perisomatic contacts was variable in both BC types, e.g. there were cells, which avoided the perisomatic region and formed synapses only on more distal sites, whereas others innervated exclusively the perisomatic membrane surface. Between these two extremes, the innervation patterns of BCs formed a continuum, therefore categorization of these interneurons into perisomatic and dendrite-targeting cells expressing PV and CCK like in the hippocampus (*Cope et al., 2002*; *Buhl et al., 1994*; *Szabó et al., 2014*) is not possible in the BA. When we analyzed the target distribution of single BCs on multiple postsynaptic cells (*Figure 8*), we found the same continuous innervation pattern, i.e. only a few percent of the BCs formed synapses with the very same distribution on all of their postsynaptic partners. What can be the reason for the high variability in innervation pattern of single BCs? Recent studies suggested that the BA PN populations show heterogeneity regarding their different afferent and efferent connections that may allow them to play distinct roles in BA functions (*Herry et al., 2008*; *Senn et al., 2014*; *Namburi et al., 2015*). These PN types are intermingled within the BA, but they might be distinctly controlled by local interneurons, as it has been shown in the hippocampus or neocortex (*Lee et al., 2014*; *Varga et al., 2010*; *Bodor et al., 2005*). However, a recent elegant study showed that PNs projecting to different prefrontal cortical areas receive uniform synaptic inhibition from CCKBCs (*Vogel et al., 2016*). Thus, in the BA, projection area of PNs might not determine the level of inhibitory control they receive from individual BCs, at least from those expressing $CB_1$ receptors. Although a similar study has not been conducted for PVBCs, the heterogeneity of the innervation pattern observed in the present work may not be primarily defined by the subtypes of PNs projecting to different regions, a hypothesis that need to be addressed in future investigations.

Our present results, together with our publication on the axo-axonic cells in the BA (*Veres et al., 2014*) showed that there are three different perisomatic region-targeting interneurons, which can control PN firing with the same efficacy, thereby they are equally in a position to shape the output of the BA. As fear learning can be bidirectionally regulated by optogenetic manipulation of the activity of PV-containing interneurons in the BA (*Wolff et al., 2014*), our data suggests that the effect of manipulation of CCKBC functions at the behavioral level might be comparable to that observed in the case of PV-expressing interneurons on the BA network output.

What can be the reason for the presence of three different perisomatic region-targeting interneuron types with the same powerful inhibitory effects on PN spiking? Several lines of evidence suggest that these interneuron types might play different roles during BA network activities and therefore in emotional information processing. For instance, how these interneuron types can be recruited by local PNs is markedly different. Morphological data (*Smith et al., 2000*) and unpublished observations from our lab showed that AACs and PVBCs are heavily innervated by local PNs, whereas CCKBCs lack the strong local feedback excitation. These results imply that AACs and PVBCs might be involved in local feedback inhibitory circuits, and shape the firing of PNs even at low or moderate

network activity levels. On the other hand, local PNs may recruit CCKBCs to provide feedback inhibition only when the BA network activity is elevated, or together with a concomitant activation of extra-amygdalar, e.g. subcortical afferents.

## Materials and methods

### Experimental animals and slice preparation

All experiments were approved by the Committee for the Scientific Ethics of Animal Research (22.1/360/3/2011) and were carried out according to the guidelines of the institutional ethical code and the Hungarian Act of Animal Care and Experimentation (1998. XXVIII. section 243/1998, renewed in 40/2013.). For recording PVBCs, transgenic mice of both sex (P18-24) expressing enhanced green fluorescent protein (eGFP) under the control of the Pvalb promoter (*Meyer et al., 2002*) were used. For targeted patching of CCKBCs, transgenic mice expressing red fluorescent protein under the control of cholecystokinin (Cck) promoter were used (BAC-CCK-DsRed) (*Máté et al., 2013*). Mice were deeply anaesthetized with isoflurane and decapitated, the brain was quickly removed and placed into ice-cold cutting solution containing (in mM): 252 sucrose, 2.5 KCl, 26 NaHCO$_3$, 0.5 CaCl$_2$, 5 MgCl$_2$, 1.25 NaH$_2$PO$_4$, 10 glucose, bubbled with 95% O$_2$/5% CO$_2$ (carbogen gas). Horizontal slices of 200 µm thickness containing the BA were prepared with a Leica VT1000S or VT1200S Vibratome (Wetzlar, Germany), and kept in an interface-type holding chamber containing artificial cerebrospinal fluid (ACSF) at 36°C that gradually cooled down to room temperature. ACSF contained (in mM) 126 NaCl, 2.5 KCl, 1.25 NaH$_2$PO$_4$, 2 MgCl$_2$, 2 CaCl$_2$, 26 NaHCO$_3$, and 10 glucose, bubbled with carbogen gas.

### Whole-cell recordings

After at least one hour of incubation, slices were transferred individually into a submerged type of recording chamber perfused with ACSF at 32 ± 2°C with a flow rate of 2–3 ml/min. Recordings were performed under visual guidance using differential interference contrast microscopy (Olympus BX61W). DsRed or eGFP were excited by a mercury arc lamp or a monochromator (Till Photonics), and the fluorescence was visualized by a CCD camera (Hamamatsu Photonics, Japan). Interneurons expressing DsRed or eGFP were randomly chosen for recordings, location of the recorded BCs in the BA are shown in *Figure 1—figure supplement 2*. Patch pipettes (4–7 MΩ) were pulled from borosilicate glass capillaries with inner filament (Hilgenberg, Germany) using a DMZ-Universal Puller (Zeitz-Instrumente GmbH, Germany). K-gluconate-based intrapipette solution used in all recordings contained (in mM): 110 K-gluconate, 4 NaCl, 2 Mg-ATP, 20 HEPES, 0,1 EGTA, 0.3 GTP (sodium salt) and 10 phosphocreatine adjusted to pH 7.3 using KOH and with an osmolarity of 290 mOsm/L. For recording the presynaptic interneurons 10 mM GABA and 0.2% biocytin were added, whereas for the postsynaptic PN 100 µM Alexa Fluor 488 hydrazide sodium salt (Invitrogen) was included. Recordings were made with a Multiclamp 700B amplifier (Molecular Devices, Foster City, CA, USA), low-pass filtered at 2 kHz, digitized at 10 kHz and recorded with in-house data acquisition and stimulus software (Stimulog, courtesy of Prof. Zoltán Nusser, Institute of Experimental Medicine, Hungarian Academy of Sciences, Budapest, Hungary). Recordings were analyzed with EVAN 1.3 (courtesy of Prof. Istvan Mody, Department of Neurology and Physiology, UCLA, CA) and Origin 9.2 (Northampton, MA). Recordings were not corrected for junction potential. To record the firing characteristics, cells were injected with 800-ms-long hyperpolarizing and depolarizing square current pulses with increasing amplitudes from 10 to 600 pA. PN identity was characterized by the broad action potential waveform, accommodating firing pattern and slow afterhyperpolarzing current as well as the post hoc morphological analysis of their spiny dendrites. For recording postsynaptic inhibitory currents (IPSCs), the presynaptic IN was held around a membrane potential of −65 mV in current clamp mode, and stimulated by brief square current pulses (2 ms, 1.5–2 nA) to evoke three action potentials at 30 Hz, and the PN was clamped at a holding potential of −40 mV. Series resistance was monitored (range: 6–20 MΩ) and compensated by 65%. To record postsynaptic inhibitory potentials (IPSPs), presynaptic cell was stimulated in the same way, and the postsynaptic PN was held in current clamp mode at around −55 mV. Bridge balance was adjusted throughout the recordings. Kinetic properties of IPSCs and IPSPs were analyzed on averaged events that were calculated excluding the transmission failures. The latency of synaptic transmission was calculated by subtracting the time of

the action potential peaks from the onset of the postsynaptic currents. The potency of inhibitory postsynaptic currents (IPSC potency) is calculated as the average amplitude of the events excluding failures.

To test the ability of BCs to inhibit PN firing, theta frequency (3.53 Hz) sinusoidal current pulses with peak-to-peak amplitudes of 30 pA and 50 pA were injected into the postsynaptic PN (see also in *Veres et al., 2014*). We chose this protocol as it has been shown that PNs in the amygdala display intrinsic membrane potential oscillations at theta frequencies in vivo, a rhythm that can facilitate theta frequency modulated periodic firing (*Pape et al., 1998*). The membrane potential of PNs was set (approximately around −55 mV) such to evoke a spike at the peak of the sinusoidal current pulses with the amplitude of 50 pA, but not of 30 pA. This adjustment maintained the membrane potential of PNs near the spiking threshold. One trial consisted of 7 sinusoidal waves (5×50 pA and 2×30 pA), repeated 10–20 times in each pair. Three action potentials at 30 Hz were evoked in the interneuron by brief square current pulses (2 ms, 1.5–2 nA) before the fourth sinusoidal wave (50 pA) in each trial. To calculate the reduction in firing probability, the firing probability of PNs under control conditions was calculated from the average of the responses to 50 pA currents (1 st, third, fifth and sixth sinusoidal wave), which was compared to that obtained during the fourth cycle. The inhibitory efficacy shows the probability of the suppression of action potential generation in the PN by BC firing (i.e.: 1- [PN firing probability during the control sinusoidal current cycles - firing probability when the BC is stimulated]). To test the ability of BCs to postpone PN firing, the timing of BC stimulation was systematically shifted with 10 ms steps relative to the peak of the sinusoidal current cycles.

## Perforated patch recordings

For perforated patch recordings the same protocol was used as published in *Veres et al. (2014)*. Briefly, the same K-gluconate based intrapipette solution was used for whole-cell recordings (see above) containing 100 µg/ml gramicidin, 100 µM Alexa 488 hydrazide sodium salt and 1 mM QX-314 (Sigma) in addition. The tip of the pipette was filled with gramicidin free solution and then back-filled with the gramicidin containing intrapipette solution. The series resistance was monitored throughout the experiment, and recordings were started when the resistance fell below 100 MΩ. Pipette capacitance was neutralized and bridge balance was carefully adjusted throughout the recordings. Patch rupture was detected by (i) the inability to evoke action potentials with depolarizing current steps (the consequence of the Na+/K+ channel blocker QX-314 diffusion into the cell), (ii) with the penetration of Alexa 488 dye into the cell, or (iii) sudden drop of the access resistance, and in such cases the experiment was terminated. To estimate the reversal potential of the evoked postsynaptic responses (*Figure 1I–K*), we plotted the area (integral) of IPSPs as a function of membrane potential and obtained the value, where the second order polynomial fit crossed the x axis. For comparing the reversal potential of IPSPs in perforated patch and whole-cell mode, we compared the responses from the same presynaptic IN onto a PN recorded in perforated patch mode and a neighboring PN recorded in whole-cell mode.

## Pharmacological manipulation of perisomatic synapses

To block GABAergic synaptic transmission locally at the perisomatic region of PNs we applied a GABA$_A$ antagonist gabazine (1 µM, Tocris Bioscience) with a standard patch pipette on the cell body of the PN by continuous pressure, during whole cell recording of a monosynaptically connected interneuron-PN pair as described above. The gabazine solution also contained 100 µM Alexa594 sodium salt (Invitrogen), therefore the size of the puffed area could be monitored visually with the excitation of the fluorophore with a monochromator (Till Photonics). To test the size of the area affected by the gabazine application we recorded and filled PNs with a fluorescent dye (Alexa488) and applied a small amount of GABA (100 µM, Sigma Aldrich, 30 ms long puffs applied with a Narishige microinjector model IM200, USA) locally on different locations along the dendrites together with the continuous somatic gabazine puffs. Data was analyzed on at least 10 control and 10 gabazine puff trials/location.

## Morphological analysis of the recorded pairs

After recordings slices were fixed in 4% paraformaldehyde (PFA) in 0.1 M phosphate buffer (PB; pH 7.4) overnight. For those slices which were processed for electron microscopy the fixative solution contained in addition 0.05% glutaraldehyde and 15% picric acid. Slices were then washed out with PB several times, and incubated in cryoprotective solution (30% sucrose in 0.1 M PB) for 2 hr, and then freeze-thawed three times above liquid nitrogen. BCs were visualized using Alexa 594 or Alexa 647-conjugated streptavidin (1:1000–1:3000, Invitrogen, Carlsbad, CA, USA), whereas PNs were labeled using rabbit anti-Alexa 488 primary antibody (1:1000, Invitrogen) and Alexa 488-conjugated goat anti-rabbit secondary antibody (1:200, Invitrogen). Amplification of the original Alexa488 signal with subsequent immunolabeling ensured that the fluorescent signal in the postsynaptic PNs had significantly higher intensity than the eGFP signal in neighboring PV-eGFP cells (*Figure 3—figure supplement 1*) which could have interfered with PN cell reconstruction. Confocal images were collected using a Nikon A1R microscope fitted with an oil immersion apochromatic lens (CFI Super Plan Fluor 20X objective, N.A. 0.45; z step size: 0.8–1 µm, xy: 0.31 µm/pixel). Based on the confocal images the postsynaptic PN was fully reconstructed in 3D with the Neurolucida 10.53 software and the putative synaptic sites from the presynaptic BC were marked. For the detailed analysis of the contacts, higher magnification images were taken with the same microscope (CFI Plan Apo VC60X Oil objective, z step size: 0.13 µm, xy: 0.08 µm/pixel). Only those recordings were used for morphological analysis where a single presynaptic interneuron was labeled in the slice. The distribution analysis of the synapses was performed with the Neurolucida Explorer software. Values were corrected for shrinkage and flattening of the tissue (x and y axis: no correction, z axis: 1.7). Three PVBC-PN and four CCKBC-PN pairs were further processed for correlated light and electron microscopic studies to confirm the presence of synaptic contacts. Biocytin in BCs was visualized using avidin-biotinylated horseradish peroxidase complex reaction (ABC; Vector Laboratories, Burlingame, CA, USA) with nickel-intensified 3,3-diaminobenzidine (DAB-Ni) giving a dark brown reaction product. Alexa 488 in PNs was revealed with biotin-conjugated goat anti-rabbit antibody, with ABC reaction visualized by (3,3-diaminobenzidine) DAB producing a brown end-product. Sections were then postfixed in 0.5% $OsO_4$ with 7% glucose, treated in 10% uranyl acetate, dehydrated in a graded series of ethanol, and embedded in epoxy resin (Durcupan; Sigma). Ultrathin sections of 60 nm thickness were cut, and contact sites, where the presynaptic axon made close appositions with the identified PN or with randomly sampled targets, were analyzed in serial sections.

To calculate the number of the boutons targeting the perisomatic region of the postsynaptic PN, we defined the border of the perisomatic region in each dendrite of the postsynaptic PN indirectly with a previously established method (*Vereczki et al., 2016*). Briefly, it has been shown that the 2.1 type potassium channel (Kv2.1) immunostaining delineates the soma and proximal dendrites of the neurons. We found that the length of the proximal dendrites belonging to the perisomatic region shows high variability, but it can be predicted by the diameter of the dendrites at their somatic origin (see *Figure 1G* in *Vereczki et al., 2016*). Thus, dendrites with a large basal diameter possess a longer proximal dendritic segment belonging functionally to the perisomatic region. Using the correlation between the dendrite diameter at its somatic origin and perisomatic region length, in our analysis for the innervation pattern of BCs recorded in the pairs we could determine if a dendritic contact arrived to the segment belonging to the perisomatic region or not by measuring the basal diameter of the host dendrite.

## Target distribution analysis of biocytin-filled interneurons

To estimate the target distribution of BCs using light microscopy, sections were incubated in mouse anti-Kv2.1 (1:1000, 75–014, Neuromab), which was visualized with Alexa488-conjugated donkey anti-mouse antibody (1:500, Molecular Probes). Sections were mounted on glass slides in Vectashield (Vector Laboratories). Images were taken using an A1R or a C2 confocal laser scanning microscope (Nikon CFI Plan Apo VC60X Oil objective, z step size: 0.13 µm, xy: 0.08 µm/pixel). To investigate the number of contacts from a presynaptic BC onto the perisomatic region, terminals on 10–20 randomly chosen neighboring Kv2.1-immunolabeled cells were analyzed. To determine the ratio of perisomatic (i.e. boutons forming close appositions with Kv2.1-labeled profiles) and dendritic (i.e. boutons apposing no Kv2.1-labeled profiles) contacts, targets of 150–200 randomly chosen biocytin-filled terminals were analyzed for each BC.

## Neurochemical characterization of the CCK-DsRed and PV-eGFP cells in the BA

Mice were deeply anaesthetized and transcardially perfused with 4% PFA dissolved in 0.1M PB or with 2% PFA in 0.2M Na-acetate buffer (pH 6.0) for 20 min without post-fixation. The brain was cut coronally into 50 μm thick sections, which then were soaked in 30% sucrose overnight and the sections were kept in cryoprotectant anti-freeze solution consisting of sucrose, ethylene glycol, distilled $H_2O$, and phosphate-buffered saline (3:3:3.1 vol ratio) at –20°C until further processing was initiated. Prior to immunostaining the cryoprotectant was washed out in 0.1M PB. To immunolabel CCK in CCK-DsRed mice (n = 3), sections were incubated in rabbit anti-CCK (1:5000, Sigma Aldrich) which was visualized with Alexa488-conjugated donkey anti-rabbit antibody (1:500, Molecular Probes). To reveal PV staining in PV-eGFP mice (n = 2), sections were incubated in rabbit anti-PV antibody (1:10.000, Swant) and goat anti-GFP antibody (1:1000, Frontier Institute) which were visualized with Cy3 conjugated donkey anti-rabbit antibody (1:500, Jackson) and Alexa488-conjugated donkey anti-goat antibody (1:500, Molecular Probes), respectively. To quantify the overlap between eGFP and DsRed expression in PV-eGFPxCCK-DsRed double transgenic mice (n = 2 mice) sections were incubated in goat anti-GFP antibody (1:1000, Frontier Institute) which was visualized with Alexa488-conjugated donkey anti-goat antibody (1:500, Molecular Probes). Sections were mounted on glass slides in Vectashield (Vector Laboratories). Images were taken using a C2 confocal laser scanning microscope (Nikon CFI Super Plan Fluor ELWD 20XC objective, N.A. 0.45; z step size: 2 μm, xy: 0.64 μm/pixel).

## Statistical analysis

For comparison of data with normal distribution according to the Shapiro-Wilk test, the Two sample T-test and ANOVA were used. For data with non-normal distribution the Mann-Whitney U test (M-W test) or Wilcoxon Signed Rank test was used. For the comparison of distributions, the two sample Kolmogorov-Smirnov test was used (K-S test). All statistics were performed using Origin 8.6 or 9.2 (Northampton, MA). Data are presented as mean ± s.e.m. unless indicated.

## Acknowledgements

This work was supported by Momentum Programme of the Hungarian Academy of Sciences (Lendület, LP2012-23) and the National Office for Research and Technology (OMFB-01678/2009) awarded to NH. We thank Profs Hannah Monyer and Gábor Szabó for providing BAC-PV-eGFP mice and BAC-CCK-DsRed mice, respectively. The authors are grateful to Erzsébet Gregori, Éva Krizsán and Gergely Vörös for their excellent technical assistance and to Attila Vikór for the critical reading of the manuscript. We also thank László Barna, the Nikon Microscopy Center at the Institute of Experimental Medicine, Nikon Austria GmbH, and Auro-Science Consulting, Ltd., for kindly providing microscopy support. The authors declare no competing financial interests.

## Additional information

### Funding

| Funder | Grant reference number | Author |
| --- | --- | --- |
| Magyar Tudományos Akadémia | LP2012-23 | Norbert Hájos |
| National Office for Research and Technology | OMFB-01678/2009 | Norbert Hájos |

The funders had no role in study design, data collection and interpretation, or the decision to submit the work for publication.

### Author contributions

JMV, Conception and design, Acquisition of data, Analysis and interpretation of data, Drafting or revising the article; GAN, Acquisition of data, Analysis and interpretation of data; NH, Conception and design, Analysis and interpretation of data, Drafting or revising the article

## Author ORCIDs

Norbert Hájos, http://orcid.org/0000-0002-4582-2708

## Ethics

Animal experimentation: All experiments were approved by the Committee for the Scientific Ethics of Animal Research (22.1/360/3/2011). All of the animals were handled according to the guidelines of the institutional ethical code and the Hungarian Act of Animal Care and Experimentation (1998. XXVIII. section 243/1998, renewed in 40/2013.), and every effort was made to minimize suffering.

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
