## [Decision Letter]

Thank you for submitting your article "Perisomatic GABAergic synapses of basket cells effectively control principal neuron activity in amygdala networks" for consideration by *eLife*. Your article has been reviewed by two peer reviewers, Andreas Lüthi (Reviewer #2) and Marco Capogna (Reviewer #3), and the evaluation has been overseen by a Reviewing Editor and Andrew King as the Senior Editor.

The reviewers have discussed the reviews with one another and the Reviewing Editor has drafted this decision to help you prepare a revised submission.

Summary:

This work compares anatomical and functional features of cholecystokinin (CCK) and parvalbumin (PV) basket cells on principal cells of the basolateral amygdala in mice. A comprehensive approach encompassing paired electrophysiological recordings, immunohistochemistry and electron microscopy yields convincing results that together support the surprising conclusion that the rules of CCK/PV connectivity to principal cells are quite distinct in amygdala compared to what is known from extensive studies in hippocampus. In particular, CCK and PV interneurons have similar functional impact onto pyramidal neurons in that they each make numerous contacts with somata, where they strongly influence spiking, as opposed to hippocampus where CCK cells have a more prominent dendrite inhibiting role.

Essential revisions:

1) The specificity of the two reporter mouse lines used in this study needs better characterization. The authors report overlaps of CB1R expression with DsRed, and of calbindin with eGFP. However, neither CB1 nor calbindin are markers for the entire population of CCK or PV interneurons, respectively. The authors should quantify the extent of overlap between DsRed/eGFP-positive and CCK/PV-expressing neurons.

2) The major finding of the manuscript is that the innervation pattern of individual basket cells display high variability and that consequently the functional impact of those cells on postsynaptic targets is heterogeneous. This is a very interesting finding that is well corroborated by the experimental data. However, the authors do not provide any mechanism for the observed variability or much discussion of the impact of this. Is there any target specificity factor? Is this variability dependent on the location of the soma of the pre- and/or postsynaptic neurons? Could the authors discuss and provide some explanations for the heterogeneity observed?

---

## [Author Response]

*Essential revisions:*

*1) The specificity of the two reporter mouse lines used in this study needs better characterization. The authors report overlaps of CB1R expression with DsRed, and of calbindin with eGFP. However, neither CB1 nor calbindin are markers for the entire population of CCK or PV interneurons, respectively. The authors should quantify the extent of overlap between DsRed/eGFP-positive and CCK/PV-expressing neurons.*

We examinedthe overlap between DsRed and CCK as well as eGFP and PV using immunostainings as the reviewers requested. Moreover, to quantify the overlap between DsRed and eGFP expressing cells, we analyzed the expression of the two fluorescent proteins in the BA in the double transgenic offspring by crossing the two mice lines. The results of these investigations are presented in Figure 1—figure supplement 1, and we briefly describe the findings in the manuscript (subsection “CCKBCs and PVBCs provide inhibitory input onto PNs with similar magnitude”, first paragraph).

*2) The major finding of the manuscript is that the innervation pattern of individual basket cells display high variability and that consequently the functional impact of those cells on postsynaptic targets is heterogeneous. This is a very interesting finding that is well corroborated by the experimental data. However, the authors do not provide any mechanism for the observed variability or much discussion of the impact of this. Is there any target specificity factor? Is this variability dependent on the location of the soma of the pre- and/or postsynaptic neurons? Could the authors discuss and provide some explanations for the heterogeneity observed?*

To explore whether there is a correlation between the number of the perisomatic contacts (using as a morphological proxy for the strength of perisomatic inhibition) and the location of the postsynaptic cells, we produced a map depicting these data using the horizontal mouse brain atlas (Figure 5—figure supplement 1). This analysis showed that postsynaptic cells receiving different levels of inhibitory control from both basket cell types are distributed in a salt-and-pepper like manner in the BA. There are no privileged areas where the number of perisomatic inhibitory contacts established by single interneurons is higher or lower, therefore, we concluded that the location of PNs is not a determinant factor of the inhibitory control. We included this data into the manuscript (subsection “CCKBCs and PVBCs inhibit PN spiking with similar efficacy”, last paragraph).

We also analyzed whether the distance between the pre- and postsynaptic cell can determine the functional impact of the inhibition, as observed in the hippocampus in a recent study (Strüber et al., 2015, PNAS). As shown in Figure 5—figure supplement 2, there is no correlation between the intersomatic distance and the total number of contacts, number of perisomatic contacts or inhibitory efficacy. Therefore, we concluded that the inhibitory impact of the presynaptic basket cells on PNs is independent of the distance between the pre- and postsynaptic cells in the BA, at least within intersomatic distance of 150 µm. We included this data into the manuscript (subsection “CCKBCs and PVBCs inhibit PN spiking with similar efficacy”, last paragraph).

The other source of this heterogeneity can be that PNs in the BA projecting to different areas might be controlled differently by local interneurons. This point has been now addressed in Discussion: “Recent studies suggested that the BA PN populations show heterogeneity regarding their different afferent and efferent connections that may allow them to play distinct roles in BA functions (Herry et al. 2008, Senn et al. 2014, Namburi et al. 2015). […] Although a similar study has not been conducted for PVBCs, the heterogeneity of the innervation pattern observed in the present work may not be primarily defined by the subtypes of PNs projecting to different regions, a hypothesis that need to be addressed in future investigations.”